# The Influence of Endurance Training on the Lipid Profile, Body Mass Composition and Cardiovascular Efficiency in Middle-Aged Cross-Country Skiers

**DOI:** 10.3390/ijerph182010928

**Published:** 2021-10-18

**Authors:** Natalia Grzebisz-Zatońska, Tomasz Grzywacz, Zbigniew Waśkiewicz

**Affiliations:** 1Faculty of Cosmetology, Warsaw College of Engineering and Health, 02-366 Warsaw, Poland; n.grzebisz@gmail.com; 2Faculty of Dietetics, Vistula School of Hospitality, 02-787 Warsaw, Poland; 3Department of Sport, Institute of Physical Culture, Kazimierz Wielki University in Bydgoszcz, Chodkiewicza 30, 85-064 Bydgoszcz, Poland; tomasz.grzywacz@ukw.edu.pl; 4Institute of Sport Science, Jerzy Kukuczka Academy of Physical Education, 40-065 Katowice, Poland; 5Department of Sports Medicine and Medical Rehabilitation Moscow, Sechenov First Moscow State Medical University, 19991 Moscow, Russia

**Keywords:** lipid profile, cross-country skiing, long distance, cardiovascular capacity, body fat mass, preparatory period, roller ski, amateur

## Abstract

Monitoring the training of amateur long-distance cross-country skiers (XCS) can help athletes’ achieve a higher exercise capacity and protect their health. The aim of this study was to assess body composition changes and lipid profiles in middle-aged amateur long-distance XCS after four months of training, including specialized roller ski training. The results of the time-to-exhaustion (TTE) test and blood tests and changes in body composition were analyzed with basic descriptive statistics: the paired Wilcoxon test was used to compare the results (initial and final). Spearman’s rank correlation coefficient (R) was used to assess the influence of various variables on maximum oxygen uptake (VO_2_max). The findings show that training of amateur long-distance XCS improved maximal oxygen uptake (*p* = 0.008) and had a positive effect on fat reduction, measured in percentages (*p* = 0.038) and in kilograms (*p* = 0.023), but did not change blood lipids or other parameters. Further research could focus on other aspects of the annual training cycle: the competition period, and women in a larger group of athletes. Training with roller skis and a cross-country skiing training machine (a specialized machine for strengthening the arms and upper body) can support health and prevent obesity, overweight, and cardiovascular disease.

## 1. Introduction

In physically active people, training’s pro-health effects are noticeable. Among these changes are improvements in cardiovascular capacity, the musculoskeletal system, body composition, and lipid metabolism [1]. Physical effort also reduces mortality due to cardiovascular disease, ischemic heart disease, and coronary artery disease and is a preventive and therapeutic factor for both obesity and overweight [2,3]. Training affects the blood lipid profile, which can lower triglyceride levels and low-density lipoprotein cholesterol (LDL-C) and raise levels of high-density lipoprotein cholesterol (HDL-C) in the blood [4,5]. Such changes can be induced by summer training of amateur long-distance XCS, including roller skiing and the use of a cross-country skiing training machine (Ercolina Upper Body Power, Quero Vas (BL), Italy), which is a device for dry preparation and imitates the upper body movement of skier. Thus far, however, there are no studies and results confirming these assumptions [6].

Physical effort should be considered as a preventive and therapeutic method, and an alternative to more expensive methods (such as pharmacological treatment, surgery, or physiotherapy) to achieve the same goals [7].

Regular exercise, e.g., training of amateurs for long-distance cross-country races, can have health benefits in terms of exercise tolerance, lipid metabolism, and obesity prevention. Ren et al. [8] reported other benefits such as improved endothelial function, inflammatory response, insulin sensitivity, autonomic regulation, and blood pressure control. It is worth noting that cross-country skiing could be a perfect alternative to running—one study showed that both activities equally improved the cardiorespiratory fitness of untrained middle-aged men [9].

Although physical activity has a positive effect on the lipid profile and reduces body fat, the WHO found that over 25% of the world population is still insufficiently active, and an increase in this figure is observed in developed countries. Among recommended physical activities is aerobic endurance combined with resistance training [10]. Recent scientific research results confirm these assumptions. According to Tanasescu and coauthors [11], total physical activity that included running, weight training, and walking resulted in reduced coronary heart disease (CHD) risk. Average exercise intensity was associated with reduced risk, independent of training volume.

The growing popularity of long-distance cross-country ski races among amateurs raises questions about the impact of increased physical activity on health. Training volumes in this group far exceed public health recommendations [12]. Negative changes, such as cardiac arrhythmias, can be noted in this group. This is important in middle-aged men, in whom the risk of death due to heart disorders is increased due to a lack of collateral circulation [13]. A high proportion of all exercise-induced cardiac events (cardiac arrhythmias) occur during marathons, especially in men 35 years and older [14,15]. In other studies, ventricular premature complexes (VPCs) were present in 33 of 37 men, and arrhythmias were significantly increased in middle-aged men during exhaustive prolonged exercise as compared to those observed during a similar period of normal daily life [16]. Therefore, it seems that overall health assessments, including cardiovascular and lipid profiles, may be effective and valuable in protecting the health of amateurs.

The risk of myocardial infarction is 50 times higher in sedentary people who suddenly start exercising compared to those who regularly exercise at a moderate or high level of exercise [17]. Hence, it is important to periodize training and adjust training loads to the body. This indicates the need for physiological and biochemical monitoring and appropriate preparation of amateurs for competition. Body composition and a proper lipid profile, with an appropriately adjusted training plan, can protect the health of XCS.

While most of the literature concerns professional cross-country skiers or those who are obese and overweight, data are missing for amateurs. Monitoring changes in body composition and lipid profiles can be a critical factor in protecting against CHD risk, and public health. It is also necessary in this group to manage overloads in training and competition. Abnormal parameters in the lipid profile and body composition may contribute to the appearance of dysfunction in the circulatory system: it is important to systematically control these indicators, especially in the group of physically active middle-aged men. The aim of this study was to assess the changes in body fat content and lipid profiles in amateur middle-aged long-distance XCS after four months of specialized training. The hypothesis was that endurance training would not affect the lipid profile, while it would probably reduce body fat mass and show positive effects on VO_2_max in middle-aged cross-country skiers. This knowledge can be used by trainers, doctors, and physiotherapists in the prevention and treatment of obesity and cardiovascular diseases, which may be conditioned by changes in these variables.

## 2. Materials and Methods

### 2.1. Participants

Sixteen well-trained amateur male XCS participated in the study. They had no experience in professional sport and did not receive any financial reward. The participants were not professional athletes, and they worked professionally and could spend up to one and a half hours daily on physical training. Their average age in the first test was 37.5 years (SD ± 6.6) and 37.9 (SD ± 6.8) in the second test. The research was carried out from May to September and was synchronized with the preparation period of the annual training cycle. During the study and the week before each test, the participants did not change their eating habits (isocaloric diet). The day before the study, they did not supplement their diet with caffeine and 48 h before the study they did not perform intensive efforts.

The inclusion criteria were as follows: (1) written consent of participants; (2) at least three long-distance cross-country ski races completed in the previous season; (3) sports medicine doctor’s consent. The participants had no diseases and were not using any medications. The exclusion criteria were: lack of consent for participation in the study, poor health (any disease occurrence), or lack of medical consent.

Participants followed the training plan, the core of which was roller skiing. Roller skiing is essential for summer training because it is a perfect substitute for cross-country skis, used widely by professional athletes. This research was conducted in accordance with the principles of the Helsinki Declaration, which obtained the approval of the Bioethics Committee of the Faculty of Human Nutrition and Consumption at Warsaw University of Life Sciences (No. 38p/2018).

### 2.2. Training Loads

During the four months, participants performed the recommended training in five energy zones: these were determined individually by measuring the concentration of lactate and ergospirometric values during the TTE test (Figure 1, Figure 2 and Figure 3). The first zone (zone I, 50–60% of HRmax) included recovery efforts such as stretching, yoga, and core stability training. In the second zone (zone II, 60–70% HRmax), the training was aimed to increase the ability to perform long-term efforts in conditions of moderate fatigue, typical for those who roller ski, cycle, or use Ercolina. These two zones use free fatty acid to produce energy and significantly increase endurance capabilities [18]. The third zone (AT zone, 70–80% HRmax) combined aerobic and anaerobic efforts [19]. In terms of intensity, the nature of the efforts in this area is the most similar to that during long-distance races [20]. The submaximal and maximal zones (Submax and Max) are based on anaerobic changes, which shape speed capability [21]. Although it is not the dominant element in training of long-distance XCS, it is a vital factor in stimulating the muscles towards maximum contractions.

The training plan included five training sessions per week. Two days were for endurance training, two days were for strength training (circuit training, which shapes endurance and strength capabilities), one training session was complementary, and two days were for rest. Their duration depended on the training period and ranged from 90 to 150 min for endurance units and from 60 to 90 min for strength training.

The participants had a training plan prepared by a licensed trainer. They were informed that its full implementation was crucial during the research. No problems with its implementation were reported.

Figure 1, Figure 2 and Figure 3 present monthly summaries of comprehensive loads in hours (running, swimming, cycling, core stability workouts, and general development exercises) and another summary of targeted loads in hours (roller skis, ski imitations, exercise on the cross-country training machine (Ercolina), a workout on the machine for strengthening the arms and upper body, and special exercises). The proportions of the various forms of targeted training were as follows: roller skis 60%, ski imitations 15%, and cross-country training machine (Ercolina) 25% (of total time spent on targeted training).

These summaries show data on training intensity and hours over the study’s four training months.

### 2.3. Anthropometric and Aerobic Capacity Measurements

Body weight and composition were measured on the Tanita MC-980 MA Plus Body Composition Analyzer (Arlington Heights, IL, USA), with an eight-point electrode system. The test was carried out with fasting before the TTE test began. Body weight, body fat percentage (expressed as a % and in kg), lean mass (muscle mass as a % and in kg), and body mass index (BMI) were measured. The study was conducted in the morning (from 8 a.m. to 10 a.m.). Participants had an empty bladder and did not eat a meal prior to measurements. Tests were carried out under standardized conditions at the Sports Diagnostics Center. The TTE test (ergospirometric test) was conducted to determine VO_2_ max (which is defined as the highest value of oxygen uptake obtained in the test while meeting the criteria of maintaining the oxygen uptake plateau for at least 30 s and exceeding the respiratory exchange ratio value of 1.10) and was performed on the HP Cosmos CPET treadmill (Nussdorf-Traunstein, Germany) and the Cosmed Quark/k4B2 (Rome, Italy) with increasing speed and incline. The test started at a speed of 6 km/h and a 0% treadmill inclination. Then, every 3 min, the speed was increased by 1 km/h, and the inclination by 1%. The test was continued until the athlete’s subjective feeling of exhaustion (to the point of volitional refusal). The VO_2_max values were defined as the highest oxygen uptake values obtained in the test. Heart rate was monitored with a Garmin ANT + heart rate monitor (Olathe, KS, USA).

### 2.4. Blood Sampling and Analysis

Fasting blood was drawn from the antecubital vein in dry tubes (fasting for 12 h) during the May and September tests, after the body composition measurement and before the TTE test. The spectrophotometric method determined the lipid profile: total cholesterol, HDL-C, and triglycerides. LDL-C was calculated by the Friedewald formula [22]. The Cobas 800 measuring device from Roche Diagnostics (Basel, Switzerland) was used for the study.

### 2.5. Statistics

Statistical calculations were performed using Statistica 13.1 software (StatSoft, Tulsa, OK, United States) for MS Windows 10. The normality of data distributions was tested using the Shapiro–Wilk W-test. The Brown–Forsythe test for homogeneity of variance was carried out. Results are expressed as median and interquartile range—IQR. The Wilcoxon test was used to compare the measurements from the first and second tests. Spearman’s rank correlation coefficient (R) was used to assess the influence of various variables on maximum oxygen uptake (VO_2_max). Statistical significance was set at *p* < 0.05 for all analyses.

## 3. Results

Comparisons of body weight and adipose tissue as well as the exercise parameters and lipid profiles of the first and second tests are shown in Table 1. Significant changes were recorded in reducing body fat (% body fat mass, *p* = 0.038, and body fat mass in kg, *p* = 0.023), and maximum oxygen uptake (VO_2_max, *p* = 0.008). There were no significant changes in HDL-C, non-HDL-C, LDL-C, triglycerides, BMI, maximum heart rate, and body weight.

### Correlations for Independent Variables

Correlations for maximum oxygen uptake can be found in Table 2. Only total body weight was negatively correlated with changes in VO_2_max (mL/kg/min) (*p* = 0.049 and correlation = −0.5278). Other variables did not significantly correlate with this parameter. The correlation concerned changes between the values before and after the training period.

## 4. Discussion

This study investigated the effect of four months of training on body composition and lipid profile in middle-aged male amateur long-distance XCS.

The training plan may contribute to better adaptation to competition efforts and to reducing the negative impact of strenuous exercise on the cardiovascular system, which must be monitored individually. It appears that the training itself, based on aerobic exercise, and despite the increased volume, has a positive effect on the body. Negative changes occur while participating in competitions, caused by intensity and inflammation [23]. These changes are usually temporary, depending on the intensity and duration of the performance, but tend to normalize after the race [24].

According to our results, the largest significant change was recorded in maximum oxygen uptake (*p* = 0.008), which is used to assess the exercise capacity of endurance athletes. Rusko [25] documented that up to the age of 20, maximum oxygen uptake increases. It is significantly influenced by long-term training of a low intensity, and it has been shown in studies that VO_2_max stabilizes later. Professional skiers, however, are able to increase it by increasing both the volume and intensity of training [25].

Statistically significant changes in maximum oxygen uptake were noted more often in amateurs than professional athletes [26]. This was largely due to the improvement in movement biomechanics and body composition. Puccinelli et al. [27] showed that body composition (as well as previous experience and aerobic capacity) is a good predictor for Olympic triathlon-distance performance in amateurs. Similar results were noted by other authors in various disciplines [28,29]. Arriel et al. [30] showed that body mass and body composition could be determinant for mountain biking performance, where body fat negatively influenced the performance of amateur mountain bikers, but the fat-free mass did not. In sprint runners, a lower body fat is correlated with better speed performance [31]. In our study, this parameter did not correlate with VO_2_max; a relationship was only found with total body weight, suggesting the value of muscle mass in endurance and strength efforts: other scientists examined these issues [32].

The increased exercise capacity noted in this study may result in improved lipid oxidation. Accordingly, it may also be the case that better trained or more efficient individuals use more plasma free fatty acids [33]. These differences may also be due to gender as well as the mode of exercise [34]. Future research may focus on monitoring the changes in men and women competing in amateur ski marathons [35].

Another factor may be a change in eating habits. Our participants only declared there were no changes in diet, based on individual supply of macro- and micro-nutrients to secure the organism’s needs. Future research should consider monitoring this variable in a larger group during year-round training. 

This study confirms the effectiveness of cross-country skiing training in reducing body fat in amateurs. A decrease in body fat content, without changes in body weight, indicates an increase in muscle mass. These changes have a positive effect on exercise capacity, measured by maximum oxygen uptake. A decrease in body fat will not, on its own, directly improve maximum oxygen uptake. An increase in lean body mass and the corresponding BMI will have such an effect. These variables, however, are interrelated. The content of adipose tissue is a component of the total body weight, which defines BMI and also indirectly influences VO_2_max. Therefore, it cannot be ignored when assessing these variables.

There were no statistically significant changes in the lipid profile in this study. In other studies, a positive effect of training on the VO_2_max value was presented. However, with a decreased content of adipose tissue, some results show statistically significant changes in lipid profiles, in contrast to our research. The improvement in VO_2_max is still an important factor—but not the only one—in reducing the CHD risk and its consequences, which was achieved in our results [36,37,38].

Other studies showed that regular physical activity reduces LDL-C and triglycerides. It also increases HDL-C which is related to increased insulin sensitivity and secretion of lipoprotein lipase [39]. The lack of changes in our research could have been a result of an appropriate initial level of lipid profiles, noted in other studies. In [40], markedly low concentrations of serum LDL-C and total cholesterol, as well as the expected high concentrations of HDL-C and low concentrations of triglycerides, were shown. The lack of changes in the value of the lipid profile in our study could also be the effect of an insufficient share of high-intensity efforts and the dominance of moderate efforts. To reduce LDL-C and triglyceride levels, the intensity of aerobic exercise must be increased [4].

Other studies combining resistance and endurance training did not show significant changes in the lipid profile. This may indicate that exercise pattern repeatability may be a major determinant of a change in lipid profiles [41]. Higher-intensity aerobic exercise and moderate-intensity resistance training show optimal efficacy in modifying this [42].

### The Limitations of This Study

Some limitations of the present study have to be acknowledged. First of all, we are aware of the relatively small group of participants. However, we would like to point out that keeping more people in 4-month regular training and diagnostic tests is extremely difficult and, additionally, would significantly increase the costs of the experiment. It is also worth carrying out such tests with amateurs using roller skis in the future. Nevertheless, the present study provides some insight into the specific changes in performance and body composition in response to the proposed summer cross-country skiing training protocol. Another issue that needs to be considered is whether another training protocol with an altered structure and volume will be as effective or even more effective than that proposed in this paper. The training protocol should also be repeated on a larger number of participants. Another limiting factor in this study may be the lack of control over changes in eating habits. Our participants only declared the lack of changes in their diets. Future research should consider monitoring this variable in a larger group during year-round training. Moreover, this study lacks a control group; thus, it cannot be stated whether the training alone leads to such adaptations compared to the daily living habits of a nonactive man. However, this study was not designed to investigate the effects of four-month XCS training compared to untrained subjects, but to study what and how big is the effect of the proposed XCS training protocol on the lipid profile, body mass composition, and cardiovascular efficiency.

## 5. Conclusions

Four-month training of sixteen well-trained male amateur long-distance XCS increased their maximum oxygen consumption (VO_2_max) and reduced their body fat mass. There were no significant changes in HDL-C, non-HDL-C, LDL-C, triglycerides, BMI, maximum heart rate, and body weight. Nevertheless, it is worth pointing to the pro-health trends that were observed in these variables. Thus, the hypothesis of this research was confirmed only in part.

Due to the relatively small sample size, further research could focus on assessing changes in other parts of the annual training cycle, e.g., the competition period, a larger group of athletes, and the inclusion of women. Other indicators such as troponin and selected hormones such as insulin should be assessed for changes in the overall analysis.

## Figures and Tables

**Figure 1 ijerph-18-10928-f001:**
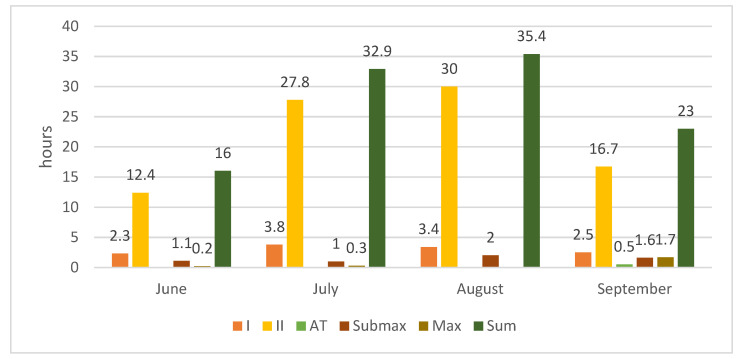
Monthly summary of comprehensive loads in hours.

**Figure 2 ijerph-18-10928-f002:**
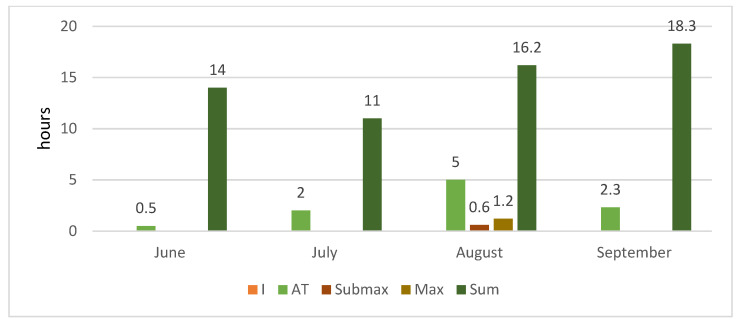
Monthly summary of targeted loads in hours.

**Figure 3 ijerph-18-10928-f003:**
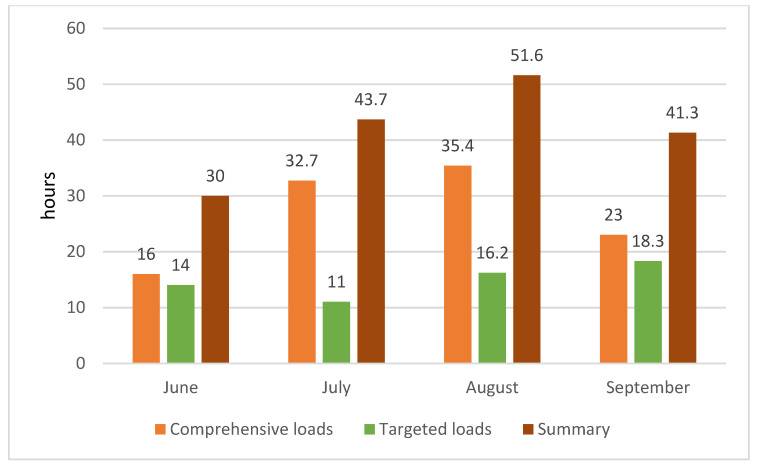
Summary of training hours.

**Table 1 ijerph-18-10928-t001:** Comparison of the first and second tests of the studied variables.

Parameter Median (Interquartile Range, IQR)	Result 1	Result 2	*p*-Value
Body weight (kg)	79.5 (7.3)	78.2 (5.9)	0.333
BMI (kg/m^2^)	23.9 (1.0)	23.9 (0.9)	0.191
% body fat mass	15.5 (3.5)	14.4 (3.4)	0.038
Body fat mass (kg)	11.1 (4.0)	10.1 (3.4)	0.023
Relative VO_2_max (mL/kg/min)	49.6 (5.0))	51.6 (4.3)	0.008
Maximum heart rate (bpm)	187.0 (7.5)	183.0 (10.0)	0.195
Cholesterol high-density lipoproteins (HDL) (mg/dL)	58.1 (13.2)	55.7 (25.5)	0.551
Cholesterol non-HDL (mg/dL)	114.2 (39.4)	112.7 (37.7)	1.000
Cholesterol low-density lipoproteins (LDL) (mg/dL)	98.7 (35.4)	98.8 (39.1)	0.233
Triglycerides (mg/dL)	72.0 (40.2)	56.0 (54.4)	0.572
Total cholesterol (mg/dL)	165.0 (38.2)	169.0 (39.1)	0.286

**Table 2 ijerph-18-10928-t002:** The *p*-values and correlations for VO_2_max.

Variable	*p*-Value	Correlation (R Spearman)
Body weight (kg)	0.049	−0.5278
BMI (kg/m^2^)	0.069	−0.4782
% body fat mass	0.880	0.0778
Body fat mass (kg)	0.858	−0.0612
Maximum heart rate (bpm)	0.138	0.4464
Cholesterol high-density lipoproteins (HDL) (mg/dL)	0.538	0.1838
Cholesterol non-HDL (mg/dL)	0.712	−0.1077
Cholesterol low-density lipoproteins (LDL) (mg/dL)	0.699	−0.1207
Triglycerides (mg/dL)	0.956	0.0201
Total cholesterol (mg/dL)	0.247	−0.3328

## Data Availability

Data are available on request due to privacy and ethical restrictions.

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
