# Peer review of "The Influence of Endurance Training on the Lipid Profile, Body Mass Composition and Cardiovascular Efficiency in Middle-Aged Cross-Country Skiers"

_ijerph, 2021, doi:10.3390/ijerph182010928_

Round 1

Reviewer 1 Report

The aim of this study was to assess body composition changes and lipid profiles in cross-country skiers after four months of training.

Although all research that helps to identify the potential of each of the physical activities for the health of their participants is interesting, this research has serious methodological errors. Moreover, its writing should be considerably improved.

Here are a some observations:

  • The keywords must be different from the title, they provide additional information to the title.
  • Many statements are made during the introduction, which should be substantiated by a reference.
  • The introduction does not follow a common thread. The ideas should be ordered, and each paragraph should lead to the next paragraph/idea. 
  • The mean age of the subjects is shown but not the deviation, it is necessary to add it.
  • Although the article focuses on the practice of XCS, the subjects perform the same XCS sessions as resistance training. It could be that resistance training alone improves their body composition and lipid profile, regardless of the endurance activity they perform. This is a significant limitation of the study.
  • Subjects performed the maximal test on an empty stomach...this may affect the results.
  • The test was performed on a treadmill, but was it performed running? When defining training intensities, what is the difference between running and XCS?
  • How was VO2max determined? add it to the methodology.
  • It is not specified what type of training methodology is used for strength.
  • Although some comments are made in the discussion, the methodology makes no reference to the nutrition maintained by the subjects during the study. Was their intake controlled? How was it controlled that they did not modify their intake?
  • Line 225. I do not doubt it, but this claim cannot be supported by this research.
  • The discussion, as well as the introduction, lacks of order and structure. The results obtained are not discussed with those of similar modalities or activities. It is necessary to discuss and value resistance training in the same way as XCS, since it is performed with the same weekly frequency.

Author Response

Thank you for your comments. See our responses in the attached file.

Reviewer 2 Report

The manuscript was greately improved, mainly in the areas of conflict of interest and discussion sectons. In this last the authors carefully describes the limitations of their results obteined in a small number of patients, and encourage further analysis.  Some new added sections in discussion have a minor spell errors. 

Author Response

Thanks for your comments, we have revised the manuscript.

Reviewer 3 Report

Thank you for the opportunity to review the manuscript. The theme of the study is relevant, however there are some important questions, regarding the methodology and the theoretical foundation of the study, that need to be answered.

Abstract: Did the authors aimed to assess cardiovascular efficiency as it has been written in the title?

Line 33, 35 and 38: These sentences need a reference

Line 45 : needs a reference

Line 48: needs a reference

Line 48: Which authors are you referring to?

Line 58: needs a reference

Line 59: needs references (The latest 58 scientific research results…).

Line 61: “The latest scientific research results” This sentence needs to be reformulated.

Line 66: Needs a reference.

Line 68: Needs a reference.

Line 63 to 75: This paragraph is out of context. It did not appear in the objectives that the study would assess the exercise negative changes.

Line 76 to 82: The sentence needs to be reformulated and the study needs to be better justified.

Line 88: Include the hypothesis

Line 93: Were the participants already well trained before the study?

Line 94: Include the standard deviation

Line 97: exclusion criteria?

Line 116: reference

Line 116: they? Who?

Line 118: reference

Line 121: Why the training was performed this way?

Figure 1, 2 and 3: Are they mean values? And SD?

Line 153: what was the VO2max assessment protocol?

What were the criteria for determining VO2max?

How was the frequency of training controlled?

Was there a change in the diet?

Were the participants already well trained?

Why expect changes in the lipid profile of individuals who were already well trained?

The study needs to be better justified.

Line 162: Data did not presented normality?

Line 167: Did you study the influence of various variables on the maximum oxygen uptake (VO2max) or the association values? There are different things.

Line 180: correlation values with the change in VO2max or with VO2max?

Line 188-202: Move to introduction section.

Line 204: As body weight has changed, it is important to present absolute values do VO2max.

Line 209-215: It is important to deeper this discussion.

Line 216 -218: reference. It is important to discuss your results about cholesterol and Triglycerides.  

Line 229: reference

Line 335: It is not a conclusion of your study.

Author Response

(The authors gave the same response as above.)

Round 2

Reviewer 1 Report

The text has been improved considerably and all doubts have been adequately answered. Congratulations.

Cross country skiers training is the most versatile of all sports. It shapes both strength, endurance and speed. Hence, its plan includes versatile, specialized and targeted units. In training amateurs in the preparatory period, it is important to shape endurance and strength abilities. This is achieved through circuit training. In this case, however, it complements the main training. The training plan also includes other activities, such as cycling. However, the goal is to achieve high exercise capacity in cross-country skiing. This is achieved not only with specialist training, but also with targeted and coprehensive training. In our opinion, it is important to pay attention not to the frequency but to the volume of training. The roller ski units last from 90 to 120 minutes. Resistance training, taking into account the time of active work (40 exercises of 30 seconds), takes about 20 minutes. For cross country skiers, both athletes and amateurs, it is a support to the main goal. Their participation and effect on the body is not commensurate, so we cannot agree.

Below we add an article analyzing the loads of the world's best cross-country skier.

Solli GS, Tønnessen E, Sandbakk Ø. The Training Characteristics of the World's Most Successful Female Cross-Country Skier. Front Physiol. 2017;8:1069. Published 2017 Dec 18. doi:10.3389/fphys.2017.01069

SANDBAKK, ØYVIND; HEGGE, ANN MAGDALEN; LOSNEGARD, THOMAS; SKATTEBO, ØYVIND; TØNNESSEN, ESPEN; HOLMBERG, HANS-CHRISTER The Physiological Capacity of the World’s Highest Ranked Female Cross-country Skiers, Medicine & Science in Sports & Exercise: June 2016 - Volume 48 - Issue 6 - p 1091-1100 doi: 10.1249/MSS.0000000000000862

It is a mistake to use literature from high-performance athletes and apply it to amateur athletes, so the article provided is of little use.

I do not question whether strength training is adequate or not for performance improvement in XCS, that is obvious, but attributing the adaptations of the training of two capacities ( resistance and endurance) to a single capacity is a serious methodological error.

Author Response

Thank you for your comment. In future studies, we will try to more clearly characterize and isolate the influence of different types of training (including endurance and resistance training) on ​​adaptation to exercise. We agree that attributing the adaptations of the training of two capacities (resistance and endurance) to a single capacity is an error. In our response, we indicated that, "This is achieved through circuit training’’. We should write that ,,this is achieved, inter alia, with circuit workout’’ or,, The latter is achieved through circuit training’’. Of course, endurance capacity is shaped by other training units, such as running, cycling, roller skis, etc.

Reviewer 3 Report

Thank you very much for answer my comments. The manuscript improved significantly. 

There is only one point that needs to be fixed. The authors answer "The VO2max values were defined as the highest  oxygen uptake values obtained in the test". This is a wrong criteria to detect VO2max. 

If there was no plateau in oxygen consumption, that is, an increase in the test speed without an increase in oxygen consumption, this last VO2 should be called  VO2peak and not VO2max. I suggest reading the article: . "Poole DC, Wilkerson DP, Jones AM. Validity of criteria for establishing
maximal O2 uptake during ramp exercise tests. Eur J Appl Physiol 102:
403–410, 2008. doi:10.1007/s00421-007-0596-3".  Please correct the sentence. 

Author Response

We changed it to:

The TTE test (ergospirometric test) was done to determine VO2 max (which is defined as the highest value ​​of oxygen uptake obtained in the test while meeting the criteria of maintaining the oxygen uptake plateau for at least 30 seconds and exceeding the respiratory exchange ratio value above 1.10), and was performed on the HP Cosmos CPET treadmill (Nussdorf-Traunstein, Germany) and the Cosmed Quark / k4B2 (Rome, Italy) with increasing speed and incline

Kind regards.

This manuscript is a resubmission of an earlier submission. The following is a list of the peer review reports and author responses from that submission.

Round 1

Reviewer 1 Report

This study was conducted to examine whether long-distance ski training can help recreational athletes improve their athletic performance and maintain their health. This study evaluated the effects of four months of training on body composition and blood biochemistry in middle-aged amateur athletes. The experimental design and subject selection for this study were not clearly described in the study methods, and the lack of clarity in the study methods for blood marker analysis may create difficulties in replicating the experiment. In addition, the design of the study and the statistical strategy were highly problematic, and the authors analyzed the correlation of different indicators using correlation methods, which mean the authors did not clearly understand the relationships of the primary outcome measurements in the field of exercise physiology. Secondly, the experimental design of this study was inappropriate because it lacked a control group of subjects for comparison and therefore it was not possible to determine whether the difference in long-distance ski training was a time effect or a training effect. Finally, the discussion logic of this study and the comprehensive comparative discussion with previous studies present incomplete information and arguments that do not help the reader to understand the main differences between this study and previous studies and do not highlight the contribution and importance of this study in the literature.

Reviewer 2 Report

The aim of this study was to assess body composition changes and lipid profiles in middle-age amateur long-distance cross-country skiers after four months of training, including specialized roller ski training, and findings showed that training of amateur long distance improved maximal oxygen uptake and had a positive effect on fat reduction. Here are my contributions: Introduction: Line 39, What other methods are you referring to? Materials and Methods: Line 109, have you practiced all these sports? of the total number of hours reported, how many hours of XCS have you done? - The lack of a control group makes the methodology worse. Discussion: Line 164 “XC skiers”, unify nomenclatures. Line 194, can this claim really be made? The limitations of the study should be added.

Reviewer 3 Report

In this report, Grzebisz-Zatońska et al., describes the observed changes on antropometric, cardiological, oxygen consumption and lipid profile obtained in male and lean Middle-Aged Cross-Country Skiers, after complete a differents trainnng programs.

The authors conclude that there were not significant changes on HDL-C, non-HDL-C, LDL-C, triglycerides, BMI, maximum heart rate, and body weight after trainning completation. However, favourable changes in body weight and % of fat mass were statistically significative.

The work-up to document BMI, cardiological and lypid profile are well described.

However, my major concern is related to include the evaluation of an apparent commercial device (named Ecorlina), which need to be punctualy described at  Introduction section (and prefereably cited properly) in order to  support/justify its evaluation into the present study. Also, as Ercolina could be considered a commercial device, any  author relationship or financial support with its fabricant (if exists) , must be clearly stated to describe any conflict of interest.

Also, I have a other comments:
- Include number of participants and specify their gender in summary section in order to support further research in women.
- The conclusion described in the following paragraph of summary, could be overestimated:
"Training with roller skis and Ercolina (specialized machine for strengthening arms 24 and upper body)
can support health and prevent obesity, overweight and cardiovascular disease."
- Paragraphs at lanes 202 to 206, and 2020 to 2021 must be interconnected to the discussion of the respective results.
- The authors could discuss about the abscense of changes on lipid profile, as all included individuals are lean (all of them have a BMI circa 23) ) and harbor normal values of cholesterol and HDL/LDL and triglycerides at the start of study?